**Data Availability Statement:** All relevant data are within the manuscript and its Supporting Information files.

# Phycocyanin attenuates skeletal muscle damage and fatigue via modulation of Nrf2 and IRS-1/AKT/mTOR pathway in exercise-induced oxidative stress in rats

Sayomphu Puengpan[1], Amnat Phetrungnapha[2], Sarawut Sattayakawee[3], Sakara Tunsophon [1,4] *

1 Department of Physiology, Faculty of Medical Science, Naresuan University, Phitsanulok, Thailand,
2 Department of Biochemistry, Faculty of Medical Science, Naresuan University, Phitsanulok, Thailand,
3 Thailand Science Research and Innovation (TSRI), Bangkok, Thailand, 4 Center of Excellence for Innovation in Chemistry, Naresuan University, Phitsanulok, Thailand

* sakarat@nu.ac.th

## Abstract

Prolonged strenuous exercise induces oxidative stress, leading to oxidative damage, skeletal muscle fatigue, and reduced exercise performance. The body compensates for oxidative stress through antioxidant actions, while related enzymes alone may not overcome excessive oxidative stress during prolonged strenuous exercise. Phycocyanin is an important antioxidant supplement derived from blue-green algae, which may be helpful in this type of situation. This study determined the effects of phycocyanin on exercise performance from prolonged strenuous exercise. Forty Sprague Dawley male rats were divided into 5 groups (n = 8 /group); Control group (C), Exercise group (E), and Exercise with supplement groups receiving low dose (Phycocyanin = 100 mg/kg BW; ELP) and high dose (Phycocyanin = 200 mg/kg BW; EHP) or vitamin C (Vitamin C = 200 mg/kg BW; VC). Phycocyanin was found to decrease oxidative damage markers, muscle fatigue, and muscle atrophy through the activated AKT/mTOR pathway. This was also found to have greater increases in antioxidants via Nrf2 signaling and increases ATP synthesis, GLUT4 transporters, and insulin signaling due to increased IRS-1/AKT signaling. In conclusion, phycocyanin was found to reduce oxidative damage and muscle atrophy, including an increase in insulin signaling in skeletal muscles leading to increased exercise performance in rats.

## Introduction

Regular exercise has a positive effect on the body. Many health studies show that exercise improves the cardiovascular system, prevents type 2 diabetes, mitigates cancer, reduces the effects of aging, and counteracts more than 40 diseases [1, 2]. Despite exercise having a positive effect on the body, prolonged high-intensity, and strenuous exercise, can also have a negative effect. Several studies have focused on the fact that excessive exercise induces oxidative stress,

**Funding:** A.P., and S.S.: Grant number 26/2561 from the Biodiversity-Based Economy Development Office (BEDO) https://www.bedo.or.th/ S.T., A.P. and S.S.: Grant number 129/2563 from the National Research Council of Thailand (NRCT) https://www.nrct.go.th/ S.T.: Grant number R2567C003, supported in part by Global and Frontier Research University Fund, Naresuan University, Thailand S.P.: Supported in part by the Center of Excellence for Innovation in Chemistry (PERCH-CIC), Ministry of Higher Education, Science, Research and Innovation, Thailand The funders had no role in study design, data collection and analysis, decision to publish, or preparation of the manuscript.

**Competing interests:** The authors have declared that no competing interests exist.

which creates negative effects on the skeletal muscles after 45 ± 5 min of swimming exercise to exhaustion in guinea pigs, or acute forced swimming exercise for 20 min, which resulted in oxidative stress in the skeletal muscles, lungs, and other organs [3, 4]. Studies have also shown that reactive oxygen species (ROS) increase significantly with intense exercise, which causes oxidative damage to the body [5].

Prolonged high-intensity and strenuous exercise can induce muscle damage because the mechanical stress experienced during exercise produces a sustained increase in ROS that activates proinflammatory cytokines, causing inflammation in the skeletal muscles which then leads to further increased ROS [1, 2]. Exercise activates nuclear factor erythroid 2-related factor 2 (Nrf2) [3], a protein that regulates antioxidant genes (*Superoxide dismutase 1 (SOD1)*, *Glutathione peroxidase 1 (GPx1)*), which helps to reduce ROS. However, in the case of prolonged vigorous exercise, the increased antioxidant activity could not overcome the dramatic increase in ROS that occurs in conjunction with the increased inflammation. This puts cells in a state of oxidative stress and leads to protein and DNA damage in skeletal muscles [2, 4]. Oxidative stress also causes abnormalities in skeletal muscle contraction in the myofilament segment, leading to decreased skeletal muscle contraction force and induced skeletal muscle atrophy. This process is mediated through the decreasing of the protein kinase B (AKT) signaling pathway which leads to increased protein degradation through the *atrogin-related transcription factors-1 (Atrogin-1)* and the *muscle RING-finger protein 1 (MuRF-1)* genes and inhibits skeletal muscle protein synthesis through the AKT/the mammalian target of rapamycin (mTOR) pathway, resulting in loss of muscle mass [5–7]. Muscle mass loss from prolonged high-intensity and strenuous exercise leads to increased muscle fatigue with a clear effect on reduced exercise performance [8].

In addition to these factors, other factors are affected by oxidative stress from prolonged high-intensity and strenuous exercise which also affects the skeletal muscles with exercise-related muscle injuries being common. In the normal state, the body can regenerate itself but in oxidative stress, ROS destroys tissues and causes mitochondria dysfunction, resulting in increased leakage of lactate dehydrogenase (LDH) and creatine kinase (CK). These destructive outcomes of ROS result in chronic muscle injury and diminish the body's ability to regenerate itself, which is one factor that reduces exercise performance [9, 10]. Exercise stimulates glucose uptake to muscle cells and glycogen storage in skeletal muscles is important for energy expenditure of the muscle. Oxidative stress also causes decreased insulin sensitivity through the interruption of the insulin receptor substrate-1 (IRS-1)/AKT pathway leading to decreased glucose transporter type 4 (GLUT4) translocation causing decreased glucose uptake in skeletal muscle [11]. The oxidative stress from prolonged high-intensity and strenuous exercise may result in reduced glucose uptake to the skeletal muscles which results in a decrease in contractile capacity due to decreased ATP production and correspondingly decreased exercise performance [12].

Phycocyanin is a natural food with blue pigment protein in Cyanobacteria, Rhodophyceae, and Cryptophyceae. While being non-toxic to normal cells, it has the potential to treat oxidative stress, with antioxidant and anti-inflammatory effects [13, 14]. Dietary supplementation with *Galdieria sulphuraria* (Rhodophyta) has been reported to reduce exercise-induced oxidative stress and mitochondrial dysfunction. Supplementation with *Galdieria sulphuraria*, which contains the macronutrients glutathione and phycocyanin, helps protect tissue from oxidative damage caused by acute exercise [15, 16]. *Gloiopeltis furcata* (Rhodophyta) contains chlorophyll-a, phycoerythrin, phycocyanin, lutein, α-carotene, β-carotene as macronutrients, which have been reported to alleviate exercise fatigue, increase energy, and increase the efficiency of swimming exercise [16].

However, there has been little research on phycocyanin to prevent exercise-induced skeletal muscle damage. Therefore, the purpose of this research was to examine the effects of phycocyanin on antioxidant activity, anti-fatigue, muscle atrophy, and its underlying mechanisms on exercise performance.

## Materials and methods

### Materials and chemicals

Phycocyanin was purchased from Xi'an Day Natural Inc. (Xi'an, Shaanxi, China). Vitamin C (L-Ascorbic acid, 99%) was purchased from Thermo Scientific Chemicals (Waltham, MA, U. S.). The glucose, LDH, and CK assay kits were purchased from Sigma-Aldrich (St. Louis, MO, USA). The assay kits for insulin, interleukin-6 (IL-6), and tumor necrosis factor alpha (TNF-α) were from EMD Millipore (Burlington, MA, USA). The chemicals for oxidative status assays were pyrogallol, glutathione, and 2-thiobarbituric acid (TBA) (Sigma-Aldrich, MO, USA), hydrogen-peroxide (Merck, MA, USA), hydrochloric acid (HCl) (RCI Labscan Ltd., Bangkok, Thailand), and trichloroacetic acid (TCA) (AppliChem GmbH, Darmstadt, Germany). The chemicals used for real-time PCR were TRI-REAGENT (Molecular Research Centre, OH, USA), reverse transcribed (RT) master mix with gDNA remover, and THUNDERBIRD™ next SYBR qPCR mix (Toyobo, Osaka, Japan). The materials and chemicals used for the western blot were RIPA lysis buffer, protease phosphatase inhibitor cocktail (Sigma-Aldrich, MO, USA), and polyvinylidene fluoride (PVDF) membrane (Merck, MA, USA). The antibodies used for Western Blot analysis were Nrf2 (Affinity Biosciences, USA), IRS-1 (Elabscience Biotechnology Co., Ltd, TX, USA), AKT (Cell Signaling, MA, USA), Phospho-AKT at Ser473 (Cell Signaling, MA, USA) and β-actin (Elabscience, TX, USA).

### Phycocyanin

The phycocyanin content was measured by the spectroscopic technique [17] as having a purity equal to 1.9 and 25% yield.

### Animal model

The animal protocols were approved by the Institutional Animal Care Committee of Naresuan University, Phitsanulok, Thailand, Ethic number: NU-AE 630605. Male Sprague Dawley rats (4–6-week-old, weighing 160–220 g) were obtained from Nomura Siam International Co., Ltd., Bangkok, Thailand, and were brought to the Center for Animal Research at Naresuan University. The rats were divided into two groups: a non-exercise group (Control or C, n = 8) and 4 exercise groups. The rats in the exercise groups were trained for swimming for one week before the experiment began and were then further divided randomly into 4 groups as follows: E Group: exercise group with no special treatment (n = 8), ELP Group: exercise + a low dose of phycocyanin 100 mg/kg BW (n = 8), EHP Group: exercise + a high dose of phycocyanin 200 mg/kg BW (n = 8), and VC Group: the positive control exercise group (Vitamin C 200 mg/kg BW, n = 8).

Phycocyanin and vitamin C were dissolved in tap water to make 1 mL of supplement that was given to the rats in daily doses of either 100 mg or 200 mg/kg BW, fed to the various groups, ELP, EHP, and VC Groups, as indicated above. The C and E Groups were treated with the same daily volume of ordinary tap water. On the eighth week, the rats were intraperitoneal injected with thiopental sodium (50 mg/kg BW, i.p.) to anesthetize. Blood was collected from the heart, and the skeletal muscles were collected immediately for muscle tension determination and muscle glucose uptake. Blood and skeletal muscle samples were kept at -80°C until

required for measurement of skeletal muscle injury and inflammation, oxidative and antioxidant status, gene and protein expressions. H&E staining was used to study the alteration of skeletal muscle the alteration of skeletal muscle structure.

## Swimming training

The exercise groups (E, ELP, EHP, VC) swam in a pool (0.50 m x 0.50 m x 0.90 m) with 30 ± 5°C water for 5 days per week for 8 weeks. In week 1 the exercising rats spent 30 min swimming per day and in week 2, they spent 40 min swimming per day. The exercise period extended to 60 min of swimming per day in weeks 3–5, and in weeks 6–8, they spent 75 min per day swimming. The animals were released to swim freely without interference. After the time had passed, the rats were scrupulously dried and returned to the housing box [18–21]. This model is described as prolonged excessive exercise according to previous studies [21, 22].

## Forced swim test (FST)

The forced swimming test (FST) for all animal groups was carried out with a weight weighing 3% of the body weight of each animal attached to the tail of the animal, which was swimming in a 45 cm x 45 cm x 45 cm pool with a height of 35 cm and a water temperature of 25°C. The animals were then timed for fatigue swimming above the water level and stopped when the animals were immobile, or the rat's nose was underwater for 5 s [23, 24]. The swimming tests were conducted at weeks 0, 2, 4, 6, and 8.

## Isolated muscle tension determination

To determine muscle contraction efficiency, the soleus muscle was used as described earlier [25, 26]. Briefly, the isolated soleus muscle (100–150 mg) was immersed in the Krebs- $Ca^{2+}$ free (KCF) buffer aerates in 95% $O_2$ and 5% $CO_2$. The soleus muscle was then attached to the impulse transducer and electrically stimulated, and the stimulation frequency was increased. The values obtained from a computer connected to the power lab and calculated for which force tension using the following muscle force (N/cm$^2$) = (force (g) x muscle length (cm) x 1.06) / (muscle weight (g) x 0.00981).

## Skeletal muscle energy and insulin level

To determine muscle uptake of glucose, the quadriceps muscle was isolated (100–150 mg) and immersed in Krebs-Ringer bicarbonate (KRB) buffer [27]. The muscle samples were then incubated and aerated in 95% $O_2$ and 5% $CO_2$ at 37°C in a KRB solution containing glucose for 5 min, then incubated for 30 min at 37°C in an insulin (50 IU/mL) and non-insulin-containing water bath. The samples were then tested with a GO assay to determine the residual glucose concentration. Glucose was expressed as mmol/dL/g muscle.

A 150 mg of the quadriceps muscle was incubated at 70°C for 30 min in 30% KOH, added 95% EtOH for 20 min on ice, and then centrifuged at 18,000 rpm for 15 min. Remove the supernatant then add D.W., 5% phenol, and 95% sulfuric acid mix. Quantification of muscle glycogen was then tested with a GO assay to determine glycogen content expressed as mg/150 mg muscle.

The fasting serum glucose and insulin levels were determined for testing glucose uptake and insulin activity in skeletal muscle. The fasting serum glucose was determined from a GO assay expressed as mg/dL, and the fasting serum insulin was determined from an ELISA kit expressed as ng/ml. Insulin sensitivity was determined with the Quantitative Insulin Sensitivity

Index (QUICKI), calculated as QUICKI = 1/[(Fasting insulin (µU/ml)) + (Fasting glucose (mg/dL)] [28].

## Skeletal muscle damage and inflammation determination by ELISA

The levels of LDH, and CK are biomarkers of muscle injury. LDH and CK were determined following the manufacturer's guidelines for using the LDH and CK commercial assay kits. The levels of LDH and CK are expressed as units/L in serum and units/mg protein in the quadriceps muscle.

The levels of inflammatory cytokines; IL-6 and TNF-α in the quadriceps muscle were determined following the manufacturer's guidelines. IL-6 and TNF-α levels are expressed as pg/mg protein.

## Oxidative stress determination

**Antioxidant enzyme activity.** Superoxide dismutase (SOD) activity was measured by inhibiting pyrogallol, which has autoxidation properties, and measured for kinetic assay every 30 s for 3 min at 420 nm by spectrophotometry. Catalase (CAT) activity was measured by a 30% change $H_2O_2$ to $2H_2O + O_2$, kinetic assay every 30 s for 5 min at 240 nm by spectrophotometer. Glutathione peroxidase (GPx) activity was measured by the reduction of GSH which has an absorbance of 412 nm. SOD and CAT activities were expressed as U/mL in serum and U/mg protein in tissue, the GPx activity was expressed as mU/mL in serum and mU/mg protein in the quadriceps muscle.

**Lipid peroxidation.** Lipid peroxidation was measured by the 2-thiobarbituric substrate reacting with the amount of malondialdehyde (MDA) in serum and the quadriceps muscle. The mixtures of 0.37% TBA, 0.25M HCL, and 15% TCA were mixed in a 1:1:1 ratio, then incubated at 95°C for 15 min, then placed on ice for 1 min to stop the reaction. The OD was read at 535 nm.

## Real-time PCR

TRI-REAGENT was used to extract RNA from the quadriceps muscle. The RNA concentration at the A260/A280 ratio was determined by NanoDrop One Spectrophotometers (Nanodrop Technologies), RT Master Mix with gDNA Remover was used to reverse transcribed RNA to cDNA at 1000 µg concentration according to the manufacturer's protocol, THUNDERBIRD™ Next SYBR qPCR Mix was used for real-time PCR. The RT-PCR conditions were incubated at 95°C for 5 min and then 40 cycles at 94°C for 30 s, with an annealing temperature of primer for 30 s, and 72°C for 30 s.

The primer sequence and annealing temperature were as follows: *SOD1* F: GCAGAAGGCAA GCGGTGAAC, R: TAGCAGGACAGCAGATGAGT, 58°C; *GPx*1 F: CTCTCCGCGGTGGCACAGT, R: CCACCACCGGGTCGGACATAC, 62°C; *mTOR* F: CAGGACGAGCGAGTGAT, R: CGAGTTGGT GGACAGAGG, 58°C; *Atrogin-1* F: AGACCGGCTACTGTGGAAGAG, R: CCGTGCATGGATCAGT G, 60°C; *MuRF-1* F: ACAACCTCTGCCGGAAGTGT, R: CCGCGGTTGGTCCAGTAG, 55°C; *SLC2A4* F: GCTTCTGTTGCCCTTCTGTC, R: TGGACGCTCTCTTTCCAACT, 60°C; *ATP5B* F: GCACCGTCAGAACTATTGCT, R: GAATTCAGGAGCCTCAGCAT, 60°C; *GAPDH* F: TGCACCACCAACTGCTTA, R: GGATGCAGGGATGATGTTC, 60°C. The relative mRNA expression was calculated through the $2^{-\Delta\Delta Ct}$ equation with *GAPDH* [29].

## Western blot analysis

The extracted protein of the individual quadriceps muscle (n = 3) was mixed into a RIPA lysis buffer and protease phosphatase inhibitor cocktail which was then homogenized and centrifuged to collect the muscle protein supernatants, 80 μg of which was loaded in sodium dodecyl sulfate-polyacrylamide gel electrophoresis and transferred to a PVDF membrane. The PVDF was blocked for nonspecific protein with 5% skimmed milk for 2 hr. to remove the skimmed milk. Primary antibodies of Nrf2, IRS-1, AKT, Phospho-AKT at Ser473, and β-actin were added to the PVDF which was then incubated at 4˚C overnight. Secondary antibodies were added to the PVDF which was incubated for 2 hr. Subsequently, the proteins were detected by Amersham TM Image Quant 800 (Cytiva, MA, USA), and analyzed with Image Lab Software 6.0.1 (Bio-rad, USA).

## Histological analysis of skeletal muscle

Quadriceps muscle tissues were embedded into a paraffin box using standard protocols. Then 5 μm cross-sections of the tissues were cut and stained with H&E staining. The cross-sectional area and diameter were visualized using a 20X power lens of a light microscope and calculated by ImageJ analysis software (US National Institutes of Health).

## Statistical analysis

All data results were expressed as mean ± standard error of the mean (SEM). The statistical analysis was a One-Way Analysis of Variance (ANOVA) followed by Tukey's multiple comparison test using GraphPad Prism 5 (GraphPad Software, Inc.). Statistical significance was considered at $p < 0.05$, $p < 0.01$ and $p < 0.001$.

# Results

## Muscle contraction and exercise endurance

In the FST test, which is a test of muscle endurance in exercise, it was observed that the swimming time began to extend in week 2 in the E, EHP, ELP, and VC groups which were able to swim for longer periods than the C group. The EHP group had a significant increase in swimming time from weeks 2–8, a time that was longer than achieved by the E group. In addition, the EHP, ELP, and VC showed less fatigue and demonstrated a significant difference in swimming time, which was also longer than the E group swimming time (Fig 1).

In the muscle force tension test, the E and ELP groups' muscle force tension significantly decreased than the C and VC groups, but non-significant changes were found in the EHP group on 10 Hz (Fig 2B). The muscle force tension in the ELP group decreased significantly more than the C and VC groups on 20 Hz and 30 Hz (Fig 2C and 2D). Therefore, it can be concluded that phycocyanin while exercising can help to increase exercise endurance but does not affect the force of muscle tension.

## Skeletal muscle injury and inflammation

Indicators of skeletal muscle injury were determined in both serum and skeletal muscle and showed increased LDH and CK levels in the E group more significantly than in the C EHP, ELP, and VC groups. In the same way, the biomarker of inflammation in skeletal muscle showed increased TNF-α and IL6 levels in the E group more significantly than in the C, EHP, ELP, and VC groups and the VC group showed an increased TNF-α levels more significantly than those experienced in the C group (Table 1).

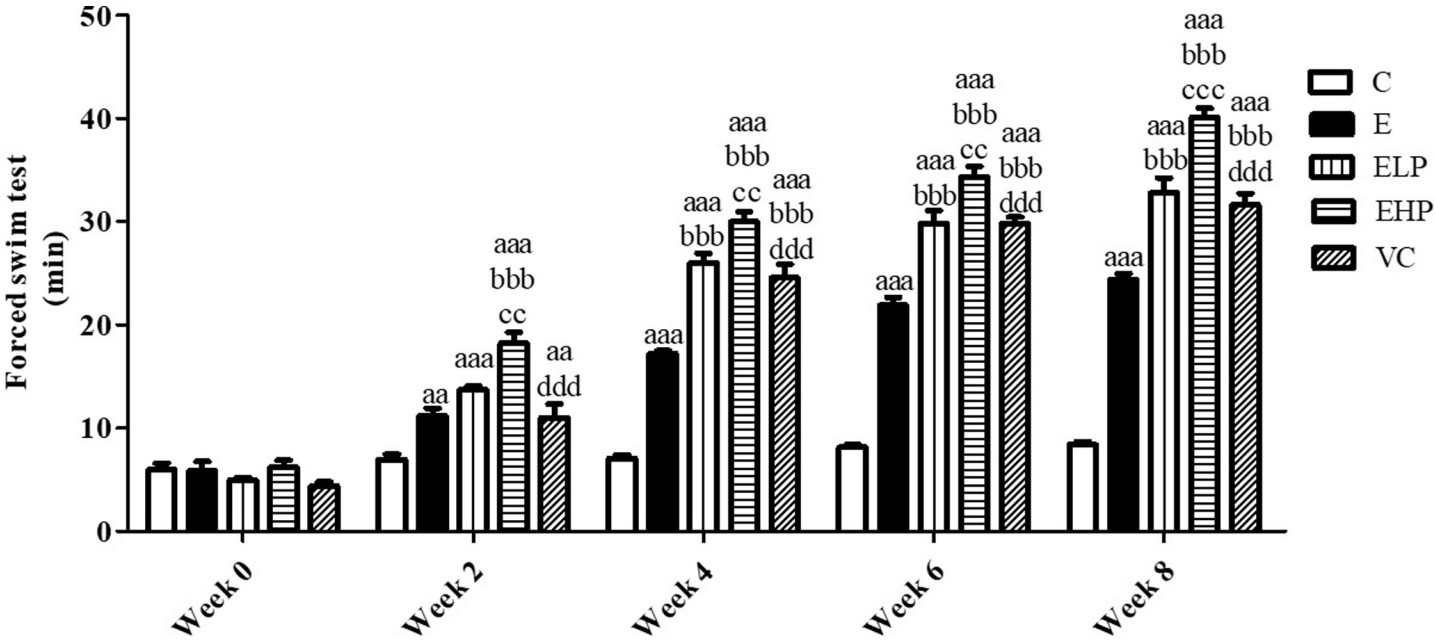

**Fig 1. Forced swim test (FST).** Swimming duration change in comparison between the C group, E group, and supplement groups to examine muscular endurance in exercise. Histograms represent mean ± S.E.M. (n = 6–8). [aa]$p<0.01$, [aaa]$p<0.001$ versus C group; [bbb]$p<0.001$ versus E group; [cc]$p<0.01$, [ccc]$p<0.001$ versus ELP group; [ddd]$p<0.001$ versus EHP group.

## Skeletal muscle energy

In order to determine muscle energy expenditure, fasting glucose, and insulin were determined. The results showed decreased levels of both fasting serum glucose and fasting serum insulin in the EHP, ELP, and VC groups. This decrease was more significant than in the C and E groups (Fig 3A and 3B). These results show that insulin sensitivity increased in the ELP,

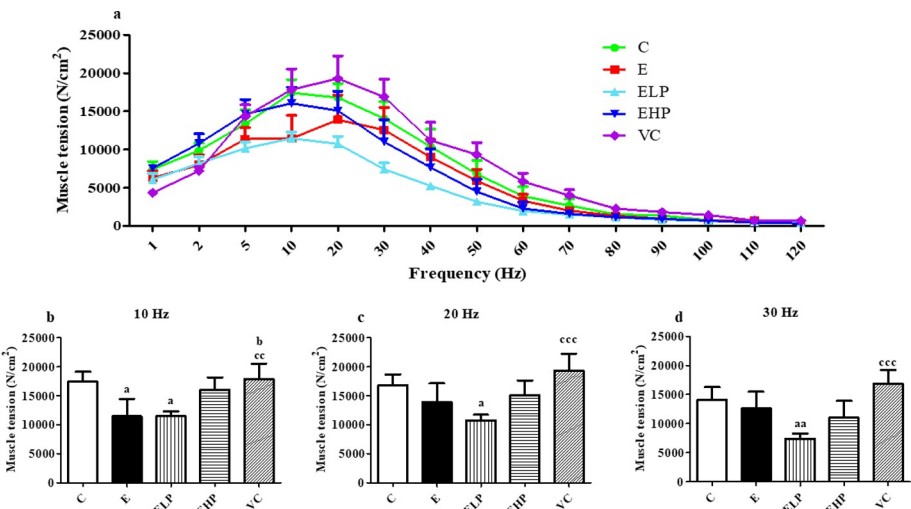

**Fig 2. Muscle force tension.** Muscle force tension analysis at the specified frequency during electrical stimulation of 10 s duration, a 20 s rest period was used. Line graphs represent all frequency and muscle tension (Mean ± S.E.M.) (a). Histograms represent some frequencies; 10 Hz (b), 20 Hz (c), 30 Hz (c) (Mean ± S.E.M., n = 3–7). [a]$p<0.05$, [aa]$p<0.01$ versus C group; [b]$p<0.05$ versus E group, [cc]$p<0.01$, [ccc]$p<0.001$ versus ELP group.

**Table 1. Parameters of body weight, muscle weight and biochemical markers for muscle injury and inflammation.**

| Parameters | Groups | | | | |
|---|---|---|---|---|---|
| | C | E | ELP | EHP | VC |
| **Body weight (g)** | 575.70±17.17 | 543.80±13.18 | 520.00±7.87[a] | 483.30±8.67[aa] | 560.40±9.20[d] |
| **Quadriceps muscle weight (g)** | 4.00±0.11 | 4.17±0.13 | 3.77±0.13 | 3.28±0.32 [a, bb] | 3.75±0.11 |
| **Quadriceps muscle mass index (mg/g)** | 6.91±0.22 | 7.67±0.26 | 7.25±0.24 | 6.49±0.49 | 6.70±0.20 |
| **Soleus muscle (g)** | 0.223±0.00 | 0.237±0.04 | 0.200±0.01 | 0.195±0.01 | 0.2043±0.01 |
| **Soleus muscle mass index (mg/g)** | 0.375±0.01 | 0.428±0.08 | 0.436±0.04 | 0.418±0.02 | 0.399±0.02 |
| **Muscle** | | | | | |
| **LDH (mU/mg protein)** | 761.80±169.70 | 1529.00±96.41[aaa] | 789.80±80.11[bbb] | 879.20±97.64[bb] | 551.60±106.40[bbb] |
| **CK (U/mg protein)** | 13.50±1.28 | 26.74±1.78[aaa] | 15.86±2.34[bb] | 13.80±3.10[bb] | 17.43±0.73[b] |
| **IL-6 (pg/mL/mg protein)** | 2024.00±1035.00 | 16230.00±3321.00[aaa] | 6180.00±746.00[bb] | 4666.00±2136.00[bb] | 1587.00±518.00[bb] |
| **TNF-α (pg/mL/mg protein)** | 24.40±5.85 | 209.00±29.04[aaa] | 33.87±12.65[bbb] | 52.57±12.76[bbb] | 104.90±26.26[a, bb] |
| **Serum** | | | | | |
| **LDH (mU/mL)** | 354.10±91.11 | 925.40±105.6[aaa] | 277.40±61.51[bbb] | 344.40±60.38[bbb] | 418.80±17.95[bbb] |
| **CK (U/L)** | 81.61±9.62 | 164.30±17.43[aaa] | 49.20±10.87[bbb] | 38.16±9.34[bbb] | 74.61±6.65[bbb] |

Data represents mean ± SEM (n = 5–8) and statistically determined by one-way ANOVA (Tukey's).

[a]$p<0.05$

[aa]$p<0.01$

[aaa]$p<0.001$ versus C group

[b]$p<0.05$

[bb]$p<0.01$

[bbb]$p<0.001$ versus E group

[d]$p<0.05$ versus EHP group.

EHP, and VC groups than in the C and E groups (Fig 3C) and the experimental results are consistent with the skeletal muscle glucose uptake test showing an increase in glucose uptake activity with insulin in the EHP, ELP, and VC groups. This increase was more significant than in the C and E groups (Fig 3D). Similarly, the glycogen content in the C and E groups decreased and the decrease was more significant than that experienced in the EHP, ELP, and VC groups (Fig 3E), while the glucose uptake activity without insulin was not significantly different in all groups (Fig 3D). In summary, EHP, ELP, and VC showed increased glucose uptake and glucose storage in the form of glycogen for reserve energy in skeletal muscle.

The experiment results showed that phycocyanin and vitamin C supplementation increased the uptake of glucose into the muscles, which modulate via the insulin signaling pathways. The predominant increased protein expressions of IRS-1 and pAKT/AKT were found in the ELP, EHP, and VC groups and considerably more than that found in the E group (Fig 4A and 4B). Moreover, the *Solute Carrier Family 2 Member 4* (*SLC2A4*) gene showed the same effect in the E group, which showed a significant decrease than in the C, EHP, ELP, and VC groups (Fig 4C) meanwhile the *mitochondrial ATP synthase F1 subunit beta* (*ATP5B*) gene, which is regulated ATP synthesis, showed a greater decrease than in the other groups (Fig 4D). Therefore, it can be concluded that phycocyanin has the effect to increased insulin signaling pathway and ATP synthesis.

## Oxidative status and antioxidant activity

Oxidative status was measured by MDA as a biomarker of oxidative stress which increased significantly in the E group, and significantly more than the increase in the C EHP, ELP, and VC

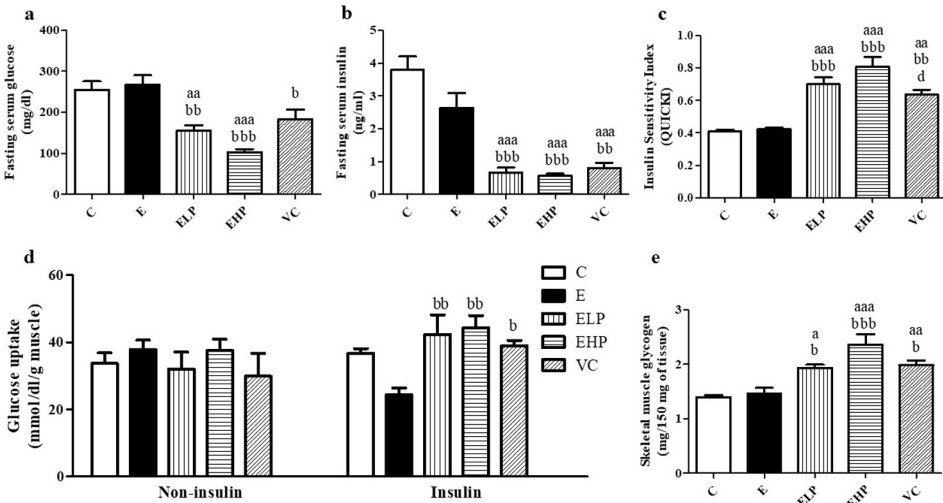

**Fig 3. Muscle glucose uptake and glycogen content.** Fasting serum glucose levels (a), Fasting serum insulin levels (b), Insulin sensitivity (c), Glucose uptake without insulin and with insulin in skeletal muscle (d), Skeletal muscle glycogen content (e), change in comparison between C group, E group, and supplement groups to examine muscle glucose activity. Histograms represent mean ± S.E.M. (n = 6–8). [a]$p<0.05$, [aa]$p<0.01$, [aaa]$p<0.001$ versus C group; [b]$p<0.05$, [bb]$p<0.01$, [bbb]$p<0.001$ versus E group.

groups. Antioxidant activity tests in both serum and muscle showed the same effect. The EHP, ELP, and VC groups increased more significantly than the increase above the E and C groups (Table 2).

The antioxidant activity was consistent with the expressions of Nrf2 by western blot analysis (Fig 5A), and expression of *SOD1* and *GPx*1 genes tested by real-time PCR (Fig 5B and 5C), the EHP, ELP, and VC groups increased more significantly than the increase in the E and C

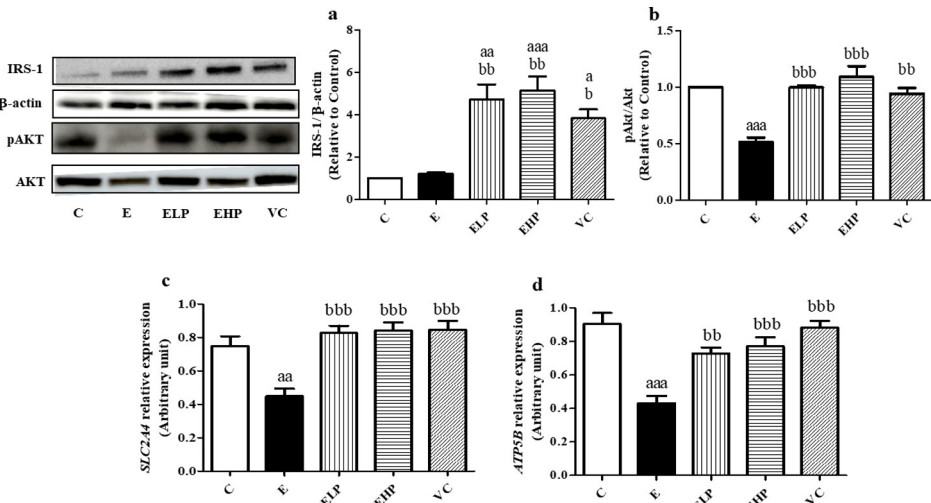

**Fig 4. Protein and gene expression of muscle energy.** Western blot analysis of IRS-1 (a.) and pAKT/AKT (b.) in skeletal muscle. Real-Time PCR *SLC2A4* gene (c.) and *ATP5B* gene in skeletal muscle (d.), change in comparison between the C group, E group and supplement groups to examine insulin signals and ATP synthesis. Histograms represent mean ± S.E.M. (n = 6–8). [a]$p<0.05$, [aa]$p<0.01$, [aaa]$p<0.001$ versus C group; [b]$p<0.05$, [bb]$p<0.01$, [bbb]$p<0.001$ versus E group.

**Table 2. The parameters of oxidative status and antioxidant enzymes in serum and muscle.**

| Parameters | Groups | | | | |
|---|---|---|---|---|---|
| | C | E | ELP | EHP | VC |
| **Muscle** | | | | | |
| MDA (μM/mg protein) | 0.22±0.04 | 0.67±0.07[aaa] | 0.12±0.01[bbb] | 0.12±0.04[bbb] | 0.17±0.03[bbb] |
| SOD (U/mg protein) | 10.49±0.64 | 12.55±0.68 | 18.18±1.05[aaa, bb] | 30.28±1.63[aaa, bbb, ccc] | 27.29±0.96[aaa, bbb, ccc] |
| CAT (U/mg protein) | 1.03±0.08 | 1.49±0.19 | 3.22±0.28[aaa, bbb] | 2.80±0.29[aaa, bb] | 2.79±0.18[aaa, bb] |
| GPx (mU/mg protein) | 508.80±38.18 | 443.90±67.60 | 1378.00±199.10[aa, bb] | 1170.00±239.90[a, b] | 1478.00±136.80[aa, bbb] |
| **Serum** | | | | | |
| MDA (μM) | 1.49±0.19 | 4.22±0.78[aa] | 1.15±0.30[bb] | 1.07±0.240[bbb] | 1.44±0.35[bbb] |
| SOD (U/min) | 124.10±11.22 | 115.90±7.77 | 189.30±8.89[aaa, bbb] | 172.70±11.69[a, bb] | 173.10±7.21[a, bb] |
| CAT (U/min) | 28.55±1.74 | 27.32±3.42 | 67.23±8.33[a, b] | 64.09±10.93[a, b] | 73.25±11.17[aa, bb] |
| GPx (mU/mL) | 1191.00±64.74 | 1292.00±102.10 | 2144.00±245.50[aa, b] | 2093.00±150.50[a] | 2529.00±105.60[aaa, bb] |

Data represents mean ± SEM (n = 6–8) and statistically determined by one-way ANOVA (Tukey's)

[a] $p < 0.05$

[aa] $p < 0.01$

[aaa] $p < 0.001$ versus C group

[b] $p < 0.05$

[bb] $p < 0.01$

[bbb] $p < 0.001$ versus E group

[ccc] $p < 0.001$ versus ELP group.

groups. It is concluded that phycocyanin showed increased antioxidant activity and decreased oxidative stress.

## Protein synthesis and protein degradation in skeletal muscle

A real-time PCR test showed that the expression of the *mTOR* gene in the E group decreased significantly more than was shown in the C, EHP, ELP, and VC groups (Fig 6A). It was also

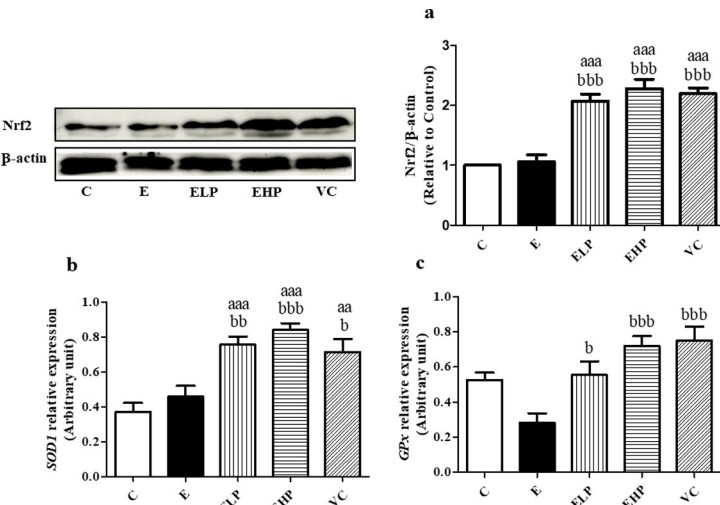

**Fig 5. Antioxidant protein and gene expressions.** Western blotting analysis of Nrf2 in skeletal muscle (a). Real-Time PCR *SOD1* gene (b) and *GPx*1 gene in skeletal muscle (c), change in comparison between C group, E group, and supplement groups to examine antioxidant activity. Histograms represent mean ± S.E.M. (n = 6–8). [aa]$p<0.01$, [aaa]$p<0.001$ versus C group; [b]$p<0.05$, [bb]$p<0.01$, [bbb]$p<0.001$ versus E group.

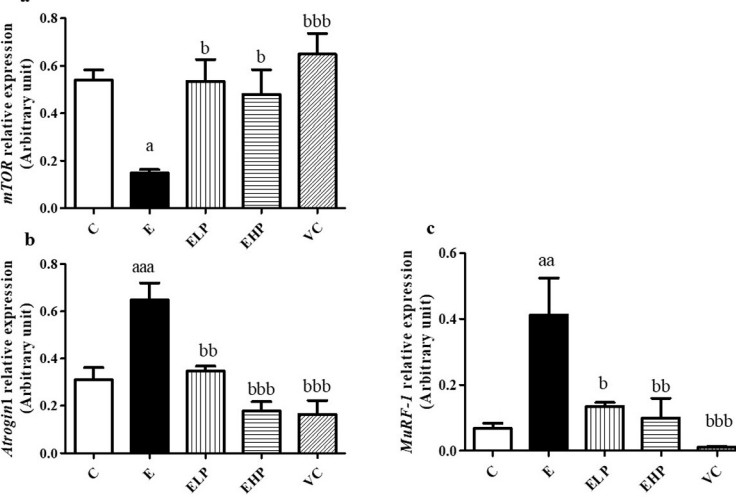

**Fig 6. Gene expressions related to protein synthesis and degradation in muscle.** The relative expression of *mTOR* (a), *Atrogin-1* (b), and *MuRF-1* (c) are normalized to *GAPDH*. RT-PCR technique was used to examine gene expression related to protein synthesis and protein degradation in skeletal muscle. Histograms represent mean ± S.E. M. (n = 6–8). [a]$p < 0.05$, [aa]$p < 0.01$, [aaa]$p < 0.001$ versus C group; [b]$p < 0.05$, [bb]$p < 0.01$, [bbb]$p < 0.001$ versus E group.

found that *Atrogin-1* and *MuRF-1* genes increased in the E group more significantly than the increase shown in the C, EHP, ELP, and VC groups (Fig 6B and 6C).

Moreover, skeletal muscle histology from H&E staining in the E group showed a greater decrease in fiber cross-sectional area (CSA) and diameter than shown in all groups (Fig 7). This result can be concluded that phycocyanin showed the potential benefit of reducing muscle atrophy.

## Discussion

Exercise is continuous movement or movement that directly affects the skeletal muscles. In high-intensity exercise, the higher energy demands require greater glucose absorbance from the bloodstream and the glycogen reserved in the skeletal muscles. This is essential to generate ATP in the mitochondria through the electron transport chain (ETC) process and also involves the release of ROS. High-intensity exercise that requires higher ATP leads to higher ROS which leads to oxidative stress. LDH and CK are cytoplasmic enzymes in skeletal and cardiac muscles. These enzymes are secreted through cell membranes that are damaged by lipid oxidation and, therefore commonly used as markers of tissue damage or injury [30, 31]. It has been reported that high-intensity exercise increased MDA levels in the specimen as a consequence of oxidative stress damage to the lipid bilayer of skeletal muscle cell membranes, resulting in the release of more CK and higher LDH levels. IL-6 and TNF-α are proinflammatory cytokines that are secreted when tissue infection, tissue injury, or oxidative stress occur, via nuclear factor kappa-light-chain-enhancer of activated B cells (NF-κB) signaling, which is a protein that regulates inflammation, immunity, and cell apoptosis in skeletal muscles. Strenuous exercise, high-intensity exercise, and resistance exercise release more IL-6 and TNF-α levels while high ROS was generated [32, 33]. Previous studies have shown that exercise-induced oxidative skeletal muscle damage can be treated with natural supplements. Spirulina supplementation for male rugby players during competitions over 7 weeks showed decreased levels of MDA which indicated a decrease in lipid peroxidation. Also, the decreased degree of CK and LDH indicated decreased muscle injury [34]. When rats were fed a diet that included *Galdieria*

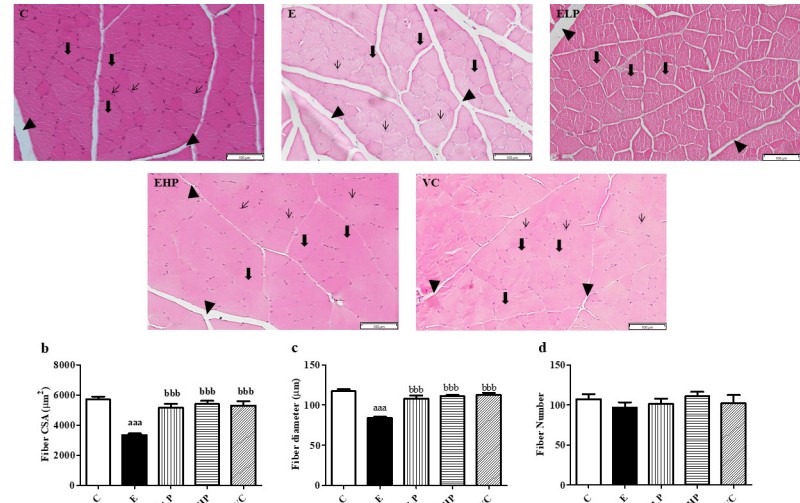

**Fig 7. Histological change by Hematoxylin and eosin stain.** Representative images were taken at a 20x magnification (Scale bar = 100 μm) of cross sections (5 μm) with H&E staining (a), the bold arrow (➡) represents muscle fiber, the arrow (→) represents the nucleus, and the arrowhead (▲) represents perimysium, change in comparison between the C group, E group, and supplement groups to examine muscle fiber size (b), fiber diameter (c) and fiber number (d) in skeletal muscle. Histograms represent mean ± S.E.M. (n = 5). [aaa]$p<0.001$ versus C group; [bbb]$p<0.001$ versus E group.

*sulphuraria* (Rhodophyta) for 10 days, then forced to swim for 6 hr. on the 10th day, and euthanized within 12 hr., they showed decreased oxidative damage in the liver, heart, and muscle [15]. Exercise to exhaustion, strenuous exercise, or chronic excessive exercise showed an increased secretion of IL-6 and TNF-α in serum [35, 36]. In our study, phycocyanin treatment reduced skeletal muscle injury and inflammation markers, demonstrating the ability of phycocyanin to repair skeletal muscle damage and inflammation which is in agreement with the prior research.

Excessive exercise or over-trained exercise was shown to be too high-intensity, or the recovery time allowed was too little, depending on the athlete's physiology, type of exercise, and other factors. Excessive exercise has been shown to result in oxidative stress, muscle damage, reduced muscle glycogen reserves, and reduced ventilatory and cardiac efficiency [37]. In our study, the E group showed extreme increases in CK, LDH, IL-6, TNF- α, and MDA levels than the increases shown in the other groups. These parameters: CK, LDH, IL-6, TNF- α, and MDA levels, are indicators of muscle damage and oxidative stress caused by excessive exercise. Additionally, our swimming training protocol was similar to a previous report of the over-training study, which increased swimming time every week [20], and swimming time over 45 min/day [38] resulted in highly increased lipid peroxidation. The swimming exercise regimen in our study is therefore considered to be excessive exercise.

Exercise has been shown to increase insulin sensitivity, increase glucose uptake, and store glycogen for energy [39]. In contrast, strenuous exercise causes oxidative stress in skeletal muscles, leading to a disturbance in the insulin signaling pathway via decreased expression of IRS-1 protein, a key part of transmitted signals in IGF-1/PI3K/AKT pathways. It is responsible for the translocation of GLUT4 in skeletal muscle to uptake glucose from the blood circulatory system and enter muscle cells [40, 41]. In our current study, the IRS-1 expression of the protein was no different in the E and C groups, which is consistent with the previous research showing decreased or no difference in IRS-1 expression from the non-exercise [40, 42]. The ELP, EHP, and VC groups have been shown to increase IRS-1 expression greater than the E group. This is consistent with the reports that phycocyanin and vitamin C have the effect of increasing the

Insulin-like growth factor 1 (IGF-1) [43, 44]. During exercise, glucose uptake in skeletal muscle is mediated by the activation of the IGF-1/IRS-1/AKT signaling pathway [45].

AKT is a key protein for major functions of glucose metabolism control, protein synthesis, and protein degradation in skeletal muscle. AKT phosphorylation via the IRS-1/PI3K signaling pathway leads to stimulating protein synthesis through the mTOR pathway, inhibiting protein degradation via the Forkhead Box O (FoxO) pathway, and translocation of GLUT4 in the skeletal muscles [46]. The previous study showed that excessive exercise-induced oxidative stress decreased phosphorylated AKT through decreased IRS-1 signaling [40, 41, 47], which is consistent with our study showed a decreased phosphorylated AKT expression in the E group compared with the other groups.

In addition, a previous study reported that the AKT activates GLUT4 translocation and increases GLUT4 synthesis by stimulating the expression of the *SLC2A4* gene [48]. Our study showed a decreased *SLC2A4* expression in the E group than other groups which is consistent with previous studies. However, phycocyanin and vitamin C showed an upregulated *SLC2A4*, resulting in increased glucose uptake and insulin sensitivity via the IRS-1/AKT pathway in skeletal muscle leading to increased glycogen content.

Exercise requires ATP for skeletal muscle contraction, using the glucose and glycogen in the skeletal muscles as a substrate. A previous study reported that excessive exercise causes mitochondrial dysfunction leading to decreased ATPase, ATP synthase, and ATP content due to ATP synthesis [49, 50]. Similarly, our study showed a decreased *ATP5B* which is responsible for ATP production in the E group than in the other groups, while phycocyanin and vitamin C supplements increase the synthesis of ATP.

Strenuous exercise has been shown to cause oxidative damage through the effects of progressively greater increases in ROS and decreased AKT activity, resulting in the activation of FoxO protein, an important regulator of *Atrogin-1* and *MuRF-1*, which are genes related to increased protein degradation in skeletal muscle [51]. In addition, the 8 weeks of strenuous exercise reduced muscle protein synthesis, by inhibiting *mTOR* which is a gene associated with protein synthesis through the inhibition of AKT, a key protein that controls protein synthesis and degradation in muscle [52]. This leads to decreased protein synthesis in skeletal muscle and results in muscle atrophy [53]. Taken together from the above information, muscle atrophy coupled with high ATP expenditure, reduced glucose uptake, and decreased glycogen storage are the causes of muscle fatigue [54]. Our study shows that the EHP significantly decreased the quadriceps muscle weight and body weight compared to the C and E groups, while the EHP group was not significant in the quadriceps muscle mass index ratio compared to the other groups. The quadriceps muscle and body weight decrease in the EHP group may be due to another pathway on exercise combined with phycocyanin. Previous studies have shown that the effect of phycocyanin improves metabolism in diabetic rats [55] and decreases body weight in obese mice [56], and exercise increases metabolism, affecting decreased body weight and body composition [57]. This may be the reason for the decreased body weight and the quadriceps muscle weight in the EHP group. However, phycocyanin showed increased protein synthesis and decreased protein degradation, which has been shown to increase muscle fiber size and prevent muscle atrophy.

The muscle force tension relationship between ATP and $Ca^{2+}$, in oxidative stress indicated that ROS inhibits ryanodine receptor type-1 (RyR1) channel function by interfering with the oxidative/nitrosative of the RyR1, which decreases $Ca^{2+}$ release from the sarcoplasmic reticulum (SR). As well, oxidative stress-induced myofilament damage or dysfunction leads to decreased muscle tension [58]. Previous studies reported that excessive exercise in rats was used as the fatigue-stimulating technique to decrease muscle tension. However, the uptake of $Ca^{2+}$ during excessive exercise showed that $Ca^{2+}$ enabled increased muscle tension. This

demonstrated that excessive exercise decreases muscle tension by interfering with $Ca^{2+}$ release in skeletal muscles [59]. In our study, the E and ELP groups showed a greater decrease in muscle tension than shown in the C and VC groups at 10 Hz, while EHP showed a good tendency to increase the tension. We can therefore assume that the damage to skeletal muscle tissue resulting from excessive exercise may interfere with $Ca^{2+}$ release in these groups. This assumption is supported by previous studies where excessive exercise was shown to interrupt $Ca^{2+}$ release. However, the tension study with phycocyanin is inconclusive, and further studies may be needed to better understand its mechanism on muscle tension.

The forced swimming test is the animal model to assess the anti-fatigue properties of the compounds. An increased exercise endurance duration manifests anti-fatigue effects and is associated with changes in several biochemical marker levels [60]. In our current study, we found that phycocyanin extract significantly increased swimming duration in the forced swim test than was observed in the control and exercise groups. Administration of phytochemicals or natural compounds has been proven to increase swim duration in animal models [61, 62]. Previous research, together with the results from our study, show that phycocyanin supplementation promotes glucose uptake through the IRS-1/AKT signaling pathway as well as increases GLUT4 translocation leading to glycogen storage and ATP synthesis, inhibiting protein degradation and increasing protein synthesis in skeletal muscles, resulting in reduced muscle fatigue. From the results of our forced swimming tests, we can conclude that phycocyanin manifests anti-fatigue effects on skeletal muscles.

As mentioned above, exercise can increase the ROS in skeletal muscles, but this is countered by the body's homeostasis through antioxidant enzymes and non-enzyme activity where ROS is converted to $H_2O$, $O_2$, and stable molecules. SOD, CAT, and GPx are antioxidant enzymes produced by the body. SOD catalyzes the superoxide anion ($O^{2.-}$) to $H_2O_2$, whereas GPx and CAT catalyze $H_2O_2$ to $H_2O$ and $O_2$. Regular low-intensity or moderate-intensity exercise in endurance and resistance training has been shown to increase antioxidant enzymes due to adaptation to decreased ROS from exercise [63, 64], whereas running for 30 min every day (non-recovery), by the rats, over 8 weeks showed decreased SOD and GPx but no change in CAT. This demonstrated that non-recovery exercise showed no change or decrease in the antioxidant enzyme activities [65]; a similar result to our study, where antioxidant enzyme activity in the E group was non-significantly different to the C group, but the ELP, EHP, and VC groups showed greater increases in antioxidant activities than in the C and E groups. Antioxidant enzymes are mostly modulated by Nrf2, thus increasing antioxidant enzymes in the ELP, EHP, and VC groups may mediate through the activation of Nrf2.

Nrf2 is a protein transcription factor of antioxidant genes (*SOD1*, *SOD2*, *Catalase*, *GPx1*) that are involved in the oxidative stress response which is activated by an increase in ROS. Exercise training has been shown to activate Nrf2 signaling which increases the antioxidant enzymes in skeletal muscle tissues in response to decreased ROS. There are conflicting reports regarding excessive exercise affecting a decrease or no change of Nrf2 in skeletal muscle, on account of deterioration, protein degradation, or damage in excessive exercise [66]. Acute exercise and endurance exercise showed an increase in Nrf2 [67], whereas exhaustive exercise showed no change in the expression of Nrf2 compared to the non-exercise group [68]. The previous study also showed that feeding with sulforaphane to the exhaustive exercise group exerts a greater increase in Nrf2 expression than the exhaustive exercise group [68]. A similar result was shown in our study, the Nrf2 expression in the E group was no different from that in the C group, while the increased Nrf2 expression was observed in the phycocyanin and vitamin C-treated groups. Specifically, exercise while being fed with phycocyanin, and vitamin C was shown to increase Nrf2 and antioxidant activities.

Vitamin C is a non-enzymatic antioxidant and functions as an electron donor to ROS. Vitamin C is currently commonly used as a supplement for athletes, which reduces exercise-induced oxidative stress for reduced fatigue, damage, and inflammation in skeletal muscle. Previous studies report that vitamin C supplementation works through the AKT/mTOR signaling pathways to reduce muscle atrophy [69], decrease inflammation and muscle pain, and oxidative stress induced by acute exercise [70]. Vitamin C increases glycogen content and glucose uptake activity in skeletal muscle through increased insulin signaling. Therefore, our study chose vitamin C as a positive control [71]. From the present study, phycocyanin treatment demonstrated equal benefits as vitamin C to reduce oxidative damage, activate antioxidant systems, improve muscle atrophy, and possess an anti-fatigue property. In addition, the human equivalence dose of phycocyanin 200 mg/kg BW in rats is equal to 1945.95 mg/60kg BW, or around 2 g/60kg BW in humans. The toxicity dose of phycocyanin in rats was reported to be greater than 3000 mg/kg BW [72], or around 29.19 g/60 kg BW in humans, so the dose in the present study is practically used and not toxic. As a result, phycocyanin has the potential to be used as an alternative supplement to reduce skeletal muscle damage in prolonged exercise.

## Conclusions

We have shown that phycocyanin works to correct muscle fatigue and reduce muscle damage associated with high-intensity exercise by reducing inflammation and oxidative stress and enhancing the activity of antioxidant enzymes via the regulation of Nrf2, *SOD1*, and *GPx*. Phycocyanin also reduces protein degradation and increases protein synthesis via *MuRF-1*, and *Atrogin-1*, *mTOR*, leading to increased muscle mass. In addition, phycocyanin increases insulin sensitivity, muscle glucose uptake, and glycogen storage through the action of the IRS-1/AKT pathway. It also increases the synthesis of ATP in the mitochondria to serve as energy while being exercised. These findings provide new knowledge regarding the reduction of

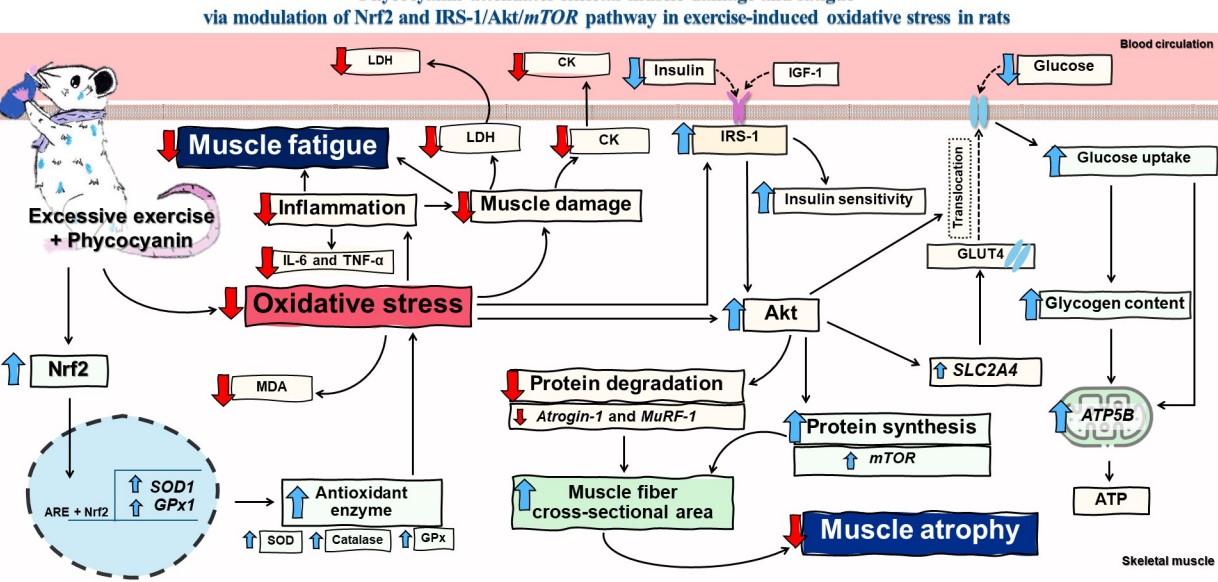

**Fig 8. Effect of phycocyanin on swimming exercise-induced oxidative stress.** Phycocyanin decreases oxidative damage and reduces inflammation due to excessive exercise, also increases the Nrf2 pathway leading to an enhancement of the antioxidant system, which causes reduced muscle damage. Phycocyanin enhances muscle glucose uptake, insulin sensitivity, GLUT4 expression, and glycogen storage, and improves protein metabolism through the IRS-1/AKT/mTOR signaling pathway. These results show that phycocyanin reduces muscle atrophy and muscle fatigue leading to improved exercise performance.

damage from high-intensity exercise and the possible mechanism is shown in Fig 8. However, its benefits for human applications should be investigated in future studies.

## Supporting information

**S1 Raw images. Raw data images of the original western blot images on the PVDF membrane, staining with immobilon forte western HRP substrate detected by Amersham TM Image Quant 800.** (DOI: 10.5281/zenodo.12749098).
(PDF)

## Acknowledgments

The author thanks Mr. Peter Barton and Mr. Kevin Mark Roebl from the International Relations and Language Development Division (DIALD) and Mr. Roy Morien of the Naresuan University Graduate School for their assistance in editing the English version of the manuscript. The author thanks Nalinnipa Wiengnak, a research assistant in the Department of Biochemistry, Faculty of Medical Science, Naresuan University for advice on the real-time PCR part, and Narongsak Pearyo from the Department of Physiology, Faculty of Medical Science, Naresuan University for advice and setting power lab machine.

## Author Contributions

**Conceptualization:** Amnat Phetrungnapha, Sarawut Sattayakawee, Sakara Tunsophon.

**Data curation:** Sayomphu Puengpan, Sakara Tunsophon.

**Formal analysis:** Sayomphu Puengpan.

**Funding acquisition:** Amnat Phetrungnapha, Sarawut Sattayakawee, Sakara Tunsophon.

**Methodology:** Sayomphu Puengpan, Sakara Tunsophon.

**Resources:** Amnat Phetrungnapha, Sarawut Sattayakawee.

**Supervision:** Amnat Phetrungnapha, Sakara Tunsophon.

**Writing – original draft:** Sayomphu Puengpan, Sakara Tunsophon.

**Writing – review & editing:** Sayomphu Puengpan, Sakara Tunsophon.

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
