## [Decision Letter · Decision Letter 0]

19 Feb 2024

PONE-D-24-00127Phycocyanin attenuates skeletal muscle damage, atrophy, and fatigue via modulation of Nrf2, mTOR, MuRF-1, and Atrogin-1 in exercise-induced oxidative stress in ratsPLOS ONE

Dear Dr. Tunsophon,

Thank you for submitting your manuscript to PLOS ONE. After careful consideration, we feel that it has merit but does not fully meet PLOS ONE’s publication criteria as it currently stands. Therefore, we invite you to submit a revised version of the manuscript that addresses the points raised during the review process.

Overall, the manuscript received positive reviews from two experts, who found it to be an interesting piece of research. However, several concerns were also raised that need to be addressed. The authors are requested to address these concerns point by point in their revised manuscript carefully. Please provide a detailed response to each comment, explaining how you have addressed the issue.

We look forward to receiving your revised manuscript.

Kind regards,

Keisuke Hitachi

Academic Editor

PLOS ONE

3. In the online submission form you indicate that your data is not available for proprietary reasons and have provided a contact point for accessing this data. Please note that your current contact point is a co-author on this manuscript. According to our Data Policy, the contact point must not be an author on the manuscript and must be an institutional contact, ideally not an individual. Please revise your data statement to a non-author institutional point of contact, such as a data access or ethics committee, and send this to us via return email. Please also include contact information for the third party organization, and please include the full citation of where the data can be found.

Reviewers' comments:

Reviewer's Responses to Questions

**Comments to the Author**

1. Is the manuscript technically sound, and do the data support the conclusions?

Reviewer #1: Yes

Reviewer #2: Yes

2. Has the statistical analysis been performed appropriately and rigorously? 

Reviewer #1: No

Reviewer #2: I Don't Know

3. Have the authors made all data underlying the findings in their manuscript fully available?

Reviewer #1: Yes

Reviewer #2: Yes

4. Is the manuscript presented in an intelligible fashion and written in standard English?

Reviewer #1: Yes

Reviewer #2: Yes

5. Review Comments to the Author

Reviewer #1: Study titled 'Phycocyanin attenuates skeletal muscle damage, atrophy, and fatigue via modulation of Nrf2, mTOR, MuRF-1, and Atrogin-1 in exercise-induced oxidative stress in rats.' It is indeed an interesting study. I have a few queries regarding the experiments and results.

1. "How was Vitamin C specifically chosen as the positive control? The rationale of choosing this. For the exercise group in the context of the experiment focusing on muscle glucose uptake and glycogen content? The authors conducted experiments on two muscle types - the abdominal muscle and the soleus muscle as can be seen in the materials and method. But used only soleus. As noted, 'To assess muscle glucose uptake, the abdominal muscle was isolated (100–150 mg) and immersed in Krebs-Ringer bicarbonate (KRB) buffer [26]. Kindly clarify the same.

2. In the endpoint please mention, which muscles have been collected.

3. Did the authors undertake an examination of fasting insulin levels in the animals as a means to evaluate insulin/glucose uptake? If so, it would be insightful to understand the baseline levels of both glucose and insulin.

4. In the analysis of NRf2 blots, an intriguing observation is the disparity between the representation in Vitamin C graphs and the actual expression in the same lane. To gain a comprehensive understanding, could the authors provide details on how many times the gel was run? Additionally, whether authors used soleus muscles were pooled or analyzed individually from each animal?

5. Given the nature of the soleus muscle as a slow-fiber muscle, it would be good if authors conduct fiber typing, preferably using techniques like COX/DSH staining or ATPase staining. This would enhance the robustness of the conclusion that phycocyanin contributes to an improvement in the observed conditions.

6. What could be the possible reasons for the increase in LDH during exercise? While there is no change in antioxidant parameters when comparing exercise and control conditions. How do the authors interpret this finding?

7. However, the tension study with phycocyanin is inconclusive, and further studies may be needed to better understand its mechanism on muscle tension. Could the author provide insights into the type of study that would be instrumental in proving the mechanism?

8. Additionally, in the table, the authors mentioned Quadriceps muscle weight (g) (MW), but all experiments utilized the soleus muscle, and the soleus muscle weight is provided. This inconsistency in muscle type raises questions about the accuracy of the reported data.

9. An intriguing observation is the maintenance of protein balance. Have the authors explored the pAKT/IGF signaling pathway or referred to any existing reports demonstrating that Phycocyanin activates the AKT/IGF signaling pathway?

10. Upon scrutiny, the histology picture presented in the panel raises concerns about the visibility of myofibers, particularly noting poor staining in panels ELP and EHP. Could the authors revise/clear on the methodology employed to calculate fiber numbers and provide a clear definition of fiber number? Furthermore, it would be valuable to include measurements of fiber diameter.

11. Regarding the statistical analysis description, there is an apparent complexity that may hinder comprehension. A thorough revision is recommended to ensure a clear and detailed explanation for better understanding.

Reviewer #2: Dear writers,

This report describes the effect of Phycocyanin against exercise-induced oxidative stress in a rat model. The results are interesting and this report seems suitable for publication in PLOS ONE. However, it should be noted that the final decision will be made by the editor. Additionally, there are some issues that need to be fixed before the article is published.

The following issues need to be reviewed.

1. Statistical analysis is not specified in the material- method section. Statistical explanations are briefly included only in the description of tables 1 and 2, however, it is not stated how the other data were evaluated statistically.

2. It was shown that both body weight and quadriceps muscle weight decreased significantly in the EHP group (table 1). This result is not discussed in the manuscript. Additionally, contradictory to these data, it was stated that phycocyanin increased protein synthesis and decreased protein degradation (lines 283, 284, 285).

3. There are abbreviation errors throughout the text. Abbreviations should be explained in full when first used in the text and then only abbreviations should be used.

4. References need to be reviewed. For example, in the 4th reference, the authors' titles are written. Volume and page numbers of some references are not written (e.g. 34, 35).

6. PLOS authors have the option to publish the peer review history of their article (what does this mean?). If published, this will include your full peer review and any attached files.

Reviewer #1: No

Reviewer #2: No

---

## [Author Response · Author response to Decision Letter 0]

17 Jul 2024

Reviewer #1: Study titled 'Phycocyanin attenuates skeletal muscle damage, atrophy, and fatigue via modulation of Nrf2, mTOR, MuRF-1, and Atrogin-1 in exercise-induced oxidative stress in rats.' It is indeed an interesting study. I have a few queries regarding the experiments and results.

Response:

We greatly appreciate the positive comment from the reviewer. 

1. "How was Vitamin C specifically chosen as the positive control? The rationale of choosing this. For the exercise group in the context of the experiment focusing on muscle glucose uptake and glycogen content? The authors conducted experiments on two muscle types - the abdominal muscle and the soleus muscle as can be seen in the materials and method. But used only soleus. As noted, 'To assess muscle glucose uptake, the abdominal muscle was isolated (100–150 mg) and immersed in Krebs-Ringer bicarbonate (KRB) buffer [26]. Kindly clarify the same.

Response: 

We are grateful that you find our research interesting. We chose vitamin C as the positive control since vitamin C or ascorbic acid is known to have high antioxidant activity, which protects the body from oxidative damage and helps strengthen the body's immunity [1]. Vitamin C is also used as a positive control in several studies related to exercise and oxidative stress [2,3] and is often used as a dietary supplement for athletes because of its strong antioxidant properties [4, 5]. Vitamin C reduces the oxidative stress effect on muscle fatigue, damage, and inflammation caused by exercise in skeletal muscles [4]. Previous studies have reported that vitamin C supplementation works effectively through the AKT/mTOR signaling pathway to reduce muscle atrophy [6], decreases muscle inflammation, muscle pain, and oxidative stress induced by acute exercise [7], and increases SR Ca2+ ATPase in the myocardium to improve myocardial contractility [8] and muscle tension [9], also increases glycogen content and glucose uptake activity in skeletal muscle through activated insulin signaling [10,11], which demonstrates a similar mechanism as phycocyanin.

For the concern regarding the muscle types, we apologize for unclear contents in the previous manuscript. We used two muscles, the soleus and quadriceps, both are muscle fiber type I [12,13]. Exercise clearly affects oxidative insulin-sensitive, glycolytic, and oxidative stress in the quadriceps muscle. Therefore, the quadriceps muscle is commonly used to test for muscle damage, glucose uptake, and insulin signaling [14,15]. The soleus muscle was used to determine the muscle force tension because of its small, and straight bundle shape. When electrically stimulated, clear changes can be seen according to the previous protocols [16-18]. We have added this information to the revised manuscript.

References 

1. Ghalibaf MHE, Kianian F, Beigoli S, Behrouz S, Marefati N, Boskabady M, et al. The effects of vitamin C on respiratory, allergic and immunological diseases: an experimental and clinical-based review. Inflammopharmacology. 2023. doi:10.1007/s10787-023-01169-1

2. Close GL, Ashton T, Cable T, Doran D, Holloway C, McArdle F, et al. Ascorbic acid supplementation does not attenuate post-exercise muscle soreness following muscle-damaging exercise but may delay the recovery process. British Journal of Nutrition. 2006;95. doi:10.1079/bjn20061732

3. Yimcharoen M, Kittikunnathum S, Suknikorn C, Nak-On W, Yeethong P, Anthony TG, et al. Effects of ascorbic acid supplementation on oxidative stress markers in healthy women following a single bout of exercise. Journal of the International Society of Sports Nutrition. 2019;16. doi:10.1186/s12970-019-0269-8

4. Evans LW, Omaye ST. Use of saliva biomarkers to monitor efficacy of vitamin C in exercise-induced oxidative stress. Antioxidants. 2017. doi:10.3390/antiox6010005

5. Hemmatfar A, Piraki P, Sharif MAS, Behpour N. Evaluating the effect of Vitamin C on myocardial angiogenesis under oxidative stress induced by exhaustive exercise in rat. Pharmaceutical Sciences. 2018;24. doi:10.15171/PS.2018.40

6. Yang M, Teng S, Ma C, Yu Y, Wang P, Yi C. Ascorbic acid inhibits senescence in mesenchymal stem cells through ROS and AKT/mTOR signaling. Cytotechnology. 2018;70. doi:10.1007/s10616-018-0220-x

7. Righi NC, Schuch FB, De Nardi AT, Pippi CM, de Almeida Righi G, Puntel GO, et al. Effects of vitamin C on oxidative stress, inflammation, muscle soreness, and strength following acute exercise: meta-analyses of randomized clinical trials. European Journal of Nutrition. 2020. doi:10.1007/s00394-020-02215-2

8. Qin F, Yan C, Patel R, Liu W, Dong E. Vitamins C and E attenuate apoptosis, β-adrenergic receptor desensitization, and sarcoplasmic reticular Ca2+ ATPase downregulation after myocardial infarction. Free Radical Biology and Medicine. 2006;40. doi:10.1016/j.freeradbiomed.2006.01.019

9. Urso ML, Clarkson PM. Oxidative stress, exercise, and antioxidant supplementation. Toxicology. 2003. doi:10.1016/S0300-483X(03)00151-3

10. Bulduk E, Gönül B, Özer Ç. Effects of vitamin C on muscle glycogen and oxidative events in experimental diabetes. Molecular and Cellular Biochemistry. 2006;292. doi:10.1007/s11010-006-9226-3

11. Picklo MJ, Thyfault JP. Vitamin e and vitamin c do not reduce insulin sensitivity but inhibit mitochondrial protein expression in exercising obese rats. Applied Physiology, Nutrition and Metabolism. 2015;40. doi:10.1139/apnm-2014-0302

12. Kriketos AD, Pan DA, Sutton JR, Hoh JFY, Baur LA, Cooney GJ, et al. Relationships between muscle membrane lipids, fiber type, and enzyme activities in sedentary and exercised rats. American Journal of Physiology - Regulatory Integrative and Comparative Physiology. 1995;269. doi:10.1152/ajpregu.1995.269.5.r1154

13. Larson L, Lioy J, Johnson J, Medler S. Transitional Hybrid Skeletal Muscle Fibers in Rat Soleus Development. Journal of Histochemistry and Cytochemistry. 2019;67. doi:10.1369/0022155419876421

 14. Yaspelkis BB, Singh MK, Trevino B, Krisan AD, Collins DE. Resistance training increases glucose uptake and transport in rat skeletal muscle. Acta Physiol Scand. 2002;175. doi:10.1046/j.1365-201X.2002.00998.x

15. Ezaki O, Higuchi M, Nakatsuka H, Kawanaka K, Itakura H. Exercise training increases glucose transporter content in skeletal muscles more efficiently from aged obese rats than young lean rats. Diabetes. 1992;41. doi:10.2337/diab.41.8.920

16. Park KH, Brotto L, Lehoang O, Brotto M, Ma J, Zhao X. Ex Vivo assessment of contractility, fatigability and alternans in isolated skeletal muscles. Journal of Visualized Experiments. 2012. doi:10.3791/4198

17. Vedsted P, Larsen AH, Madsen K, Sjøgaard G. Muscle performance following fatigue induced by isotonic and quasi-isometric contractions of rat extensor digitorum longus and soleus muscles in vitro. Acta Physiol Scand. 2003;178. doi:10.1046/j.1365-201X.2003.01123.x

18. Gomes ARS, Coutinho EL, França CN, Polonio J, Salvini TF. Effect of one stretch a week applied to the immobilized soleus muscle on rat muscle fiber morphology. Brazilian Journal of Medical and Biological Research. 2004;37. doi:10.1590/S0100-879X2004001000005

2. In the endpoint please mention, which muscles have been collected.

Response:

We apologize for the missing information. We have added details for the collected skeletal muscles under the material and method section. In brief, we used the soleus muscle for tension experiment, while the quadriceps were used in the remaining experiments.

3. Did the authors undertake an examination of fasting insulin levels in the animals as a means to evaluate insulin/glucose uptake? If so, it would be insightful to understand the baseline levels of both glucose and insulin. 

Response:

Thank you for your suggestion. Previous studies reported that excessive exercise-induced oxidative stress decreases insulin sensitivity and glucose uptake activity through decreased IRS-1 signaling [1,2], while nutrition supplementation in these conditions increases insulin sensitivity through increased IRS-1/PI3K/AKT signaling [3-5]. We therefore performed additional experiments to determine plasma insulin and glucose, and also the expression of the insulin receptor substrate-1 (IRS-1). Our study shows an increase in insulin sensitivity using QUICKI assessment [6] and increased glucose uptake activity through activation of the IRS-1 signaling pathway in excessive exercise combined with phycocyanin. These results helped us confirm insulin action in our study. We have added the new experiments and results in the revised MS.

References

1. Turgut M, Cinar V, Pala R, Tuzcu M, Orhan C, Telceken H, et al. Biotin and chromium histidinate improve glucose metabolism and proteins expression levels of IRS-1, PPAR-γ, and NF-ΚB in exercise-trained rats. Journal of the International Society of Sports Nutrition. 2018;15. doi:10.1186/s12970-018-0249-4

2. Chan SH, Kikkawa U, Matsuzaki H, Chen JH, Chang WC. Insulin receptor substrate-1 prevents autophagy-dependent cell death caused by oxidative stress in mouse NIH/3T3 cells. Journal of Biomedical Science. 2012;19. doi:10.1186/1423-0127-19-64

3. Wang H, Wang J, Zhu Y, Yan H, Lu Y. Effects of different intensity exercise on glucose metabolism and hepatic IRS/PI3K/AKT pathway in sd rats exposed with TCDD. International Journal of Environmental Research and Public Health. 2021;18. doi:10.3390/ijerph182413141

4. Kerksick CM, Wilborn CD, Roberts MD, Smith-Ryan A, Kleiner SM, Jäger R, et al. ISSN exercise & sports nutrition review update: Research & recommendations. Journal of the International Society of Sports Nutrition. 2018. doi:10.1186/s12970-018-0242-y

5. Turgut M, Cinar V, Pala R, Tuzcu M, Orhan C, Telceken H, et al. Biotin and chromium histidinate improve glucose metabolism and proteins expression levels of IRS-1, PPAR-γ, and NF-ΚB in exercise-trained rats. Journal of the International Society of Sports Nutrition. 2018;15. doi:10.1186/s12970-018-0249-4

6. Quon MJ. QUICKI Is a Useful and Accurate Index of Insulin Sensitivity. Journal of Clinical Endocrinology & Metabolism. 2002;87. doi:10.1210/jc.87.2.949

4. In the analysis of NRf2 blots, an intriguing observation is the disparity between the representation in Vitamin C graphs and the actual expression in the same lane. To gain a comprehensive understanding, could the authors provide details on how many times the gel was run? Additionally, whether authors used soleus muscles were pooled or analyzed individually from each animal?

Response:

Thank you for your comment. We have performed an additional Nrf2 western blotting and corrected the inconsistency between the NRf2 expression and histogram. By choosing the western blot image that represents the histogram as shown in Fig.5. 

We have also added additional detail on the western blot as suggested. The quadriceps were used to determine the expression of proteins of interest. We used 3 individual muscle samples/groups.

5. Given the nature of the soleus muscle as a slow-fiber muscle, it would be good if authors conduct fiber typing, preferably using techniques like COX/DSH staining or ATPase staining. This would enhance the robustness of the conclusion that phycocyanin contributes to an improvement in the observed conditions.

Response:

We appreciate your comment. We decided to perform an additional experiment to determine whether phycocyanin has any effect on ATP or not. We used the real-time PCR technique to study the gene expression of ATP synthase (Mitochondrial ATP synthase F1 subunit beta (ATP5B)). Although, this is not to confirm the muscle type, but it can provide an insightful mechanism of the PC extraction on energy expenditure by muscle cells. A previous study reported that decreased ATPase, ATP synthase, and ATP content lead to decreased ATP synthesis in excessive exercise [1,2]. Our additional result showed that the ATP5B was activated which implies that phycocyanin can stimulate the ATP synthase. This is required for ATP production during exercise. The finding contributes to the beneficial effect of phycocyanin to improve the condition of exercise-induced oxidative stress.

References

1. Tarini VAF, Carnevali LC, Arida RM, Cunha CA, Alves ES, Seeleander MCL, et al. Effect of exhaustive ultra-endurance exercise in muscular Glycogen and both Alpha1 and Alpha2 AMPK protein expression in trained rats. Journal of Physiology and Biochemistry. 2013;69. doi:10.1007/s13105-012-0224-5

2. Schluter JM, Fitts RH. Shortening velocity and ATPase activity of rat skeletal muscle fibers: Effects of endurance exercise training. American Journal of Physiology-Cell Physiology. 1994;266. doi:10.1152/ajpcell.1994.266.6.c1699

6. What could be the possible reasons for the increase in LDH during exercise? While there is no change in antioxidant parameters when comparing exercise and control conditions. How do the authors interpret this finding?

Response:

Thank you for your question. Lactate dehydrogenase (LDH) is an enzyme that catalyzes the exchange between pyruvate and lactate in cells. The membrane lipid bilayer prevents LDH from leaking out to the extracellular compartment. However, exercise-induced oxidative stress results in membrane lipid peroxidation at the membrane lipid bilayer, leading to membrane dysfunction. Therefore, the leakage of LDH can be observed in the exercise group as shown in our study. LDH is a marker used to assess tissue damage or injury [1-3]. 

A previous study showed that decreased or no change in antioxidant activity due to excessive exercise-induced oxidative stress, while increased MDA was observed in excessive exercise than non-exercise [4]. These reports are similar to our study to show that antioxidants do not change in the exercise group, but an elevated lipid peroxidation was significantly determined when compared with the other groups (C, ELP, EHP, and VC groups). Therefore, increased lipid peroxidation in the excessive exercise results in an increased leaking of LDH due to lipid membrane dysfunction.

References

1. Olayeriju OS, Olaleye MT, Crown OO, Komolafe K, Boligon AA, Athayde ML, et al. Ethylacetate extract of red onion (Allium cepa L.) tunic affects hemodynamic parameters in rats. Food Science and Human Wellness. 2015;4. doi:10.1016/j.fshw.2015.07.002

2. Tie S, Zhang L, Li B, Xing S, Wang H, Chen Y, et al. Effect of dual targeting procyanidins nanoparticles on metabolomics of lipopolysaccharide-stimulated inflammatory macrophages. Food Science and Human Wellness. 2023;12. doi:10.1016/j.fshw.2023.03.045

3. Papadopoulos NM, Leon AS, Bloor CM. Effects of Exercise on Plasma and Tissue Levels of Lactate Dehydrogenase and Isoenzymes in Rats. Proceedings of the Society for Experimental Biology and Medicine. 1967;125. doi:10.3181/00379727-125-32260

4. Wang Y, Xiang Y, Wang R, Li X, Wang J, Yu S, et al. Sulforaphane enhances Nrf2-mediated antioxidant responses of skeletal muscle induced by exhaustive exercise in HIIT mice. Food Science and Human Wellness. 2022;11. doi:10.1016/j.fshw.2022.04.035

7. However, the tension study with phycocyanin is inconclusive, and further studies may be needed to better understand its mechanism on muscle tension. Could the author provide insights into the type of study that would be instrumental in proving the mechanism?

Response:

Thank you for your question. Since muscle contraction requires ATP and Ca2+ [1], if we could perform a future experiment, we would study the ryanodine receptor type 1 (RYR1). This is because RYR1 serves as the skeletal muscle calcium release channel [2]. If the expression of RYR1 increases, it indicates increased calcium release in the skeletal muscles. An optional experiment would be the study of the Ca2+ concentration. If Ca2+ is released in greater quantities, it leads to increased muscle contraction [3].

References

1. Adams RJ, Schwartz A. Comparative mechanisms for contraction of cardiac and skeletal muscle. Chest. 1980. doi:10.1378/chest.78.1.123

2. Lawal TA, Todd JJ, Meilleur KG. Ryanodine Receptor 1-Related Myopathies: Diagnostic and Therapeutic Approaches. Neurotherapeutics. 2018. doi:10.1007/s13311-018-00677-1

3. Martins AS, Shkryl VM, Nowycky MC, Shirokova N. Reactive oxygen species contribute to Ca2+ signals produced by osmotic stress in mouse skeletal muscle fibres. Journal of Physiology. 2008;586. doi:10.111

---

## [Decision Letter · Decision Letter 1]

31 Jul 2024

PONE-D-24-00127R1Phycocyanin attenuates skeletal muscle damage and fatigue via modulation of Nrf2 and IRS-1/AKT/mTOR pathway in exercise-induced oxidative stress in ratsPLOS ONE

Dear Dr. Tunsophon,

Thank you for submitting your manuscript to PLOS ONE. After careful consideration, we feel that it has merit but does not fully meet PLOS ONE’s publication criteria as it currently stands. Therefore, we invite you to submit a revised version of the manuscript that addresses the points raised during the review process.

Your manuscript was evaluated by the two original reviewers. Before formal acceptance, minor revisions are required according to the comments by Reviewer #1. 

We look forward to receiving your revised manuscript.

Kind regards,

Keisuke Hitachi

Academic Editor

PLOS ONE

Journal Requirements:

Reviewers' comments:

Reviewer's Responses to Questions

**Comments to the Author**

1. If the authors have adequately addressed your comments raised in a previous round of review and you feel that this manuscript is now acceptable for publication, you may indicate that here to bypass the “Comments to the Author” section, enter your conflict of interest statement in the “Confidential to Editor” section, and submit your "Accept" recommendation.

Reviewer #1: All comments have been addressed

Reviewer #2: All comments have been addressed

2. Is the manuscript technically sound, and do the data support the conclusions?

Reviewer #1: Yes

Reviewer #2: Yes

3. Has the statistical analysis been performed appropriately and rigorously? 

Reviewer #1: Yes

Reviewer #2: Yes

4. Have the authors made all data underlying the findings in their manuscript fully available?

Reviewer #1: Yes

Reviewer #2: (No Response)

5. Is the manuscript presented in an intelligible fashion and written in standard English?

Reviewer #1: Yes

Reviewer #2: Yes

6. Review Comments to the Author

Reviewer #1: To enhance the clarity and consistency of the manuscript, I recommend a few minor revisions. Firstly, consolidate the information regarding the company names of chemicals and kits used throughout the manuscript into a single section within the Materials and Methods, specifically where the phycocyanin materials are described. This will streamline the text and improve readability. Secondly, for the histological images, remove the freehand circles and only arrows are sufficient. Ensure each arrow is clearly labeled to indicate its significance, thus providing precise and comprehensible visual data. Lastly, correct the spelling error in the graph of Figure 2, specifically the term "muscle tension," and ensure that all syntactical elements are accurate. These adjustments will significantly enhance the overall quality and professionalism of the manuscript. Thank you to the authors for revising the manuscript as per the reviewer’s suggestions.

Reviewer #2: (No Response)

7. PLOS authors have the option to publish the peer review history of their article (what does this mean?). If published, this will include your full peer review and any attached files.

Reviewer #1: No

Reviewer #2: No

---

## [Author Response · Author response to Decision Letter 1]

22 Aug 2024

Comments to the Author

1. If the authors have adequately addressed your comments raised in a previous round of review and you feel that this manuscript is now acceptable for publication, you may indicate that here to bypass the “Comments to the Author” section, enter your conflict of interest statement in the “Confidential to Editor” section, and submit your "Accept" recommendation.

Reviewer #1: All comments have been addressed

Reviewer #2: All comments have been addressed

2. Is the manuscript technically sound, and do the data support the conclusions?

Reviewer #1: Yes

Reviewer #2: Yes

3. Has the statistical analysis been performed appropriately and rigorously?

Reviewer #1: Yes

Reviewer #2: Yes

4. Have the authors made all data underlying the findings in their manuscript fully available?

Reviewer #1: Yes

Reviewer #2: (No Response)

5. Is the manuscript presented in an intelligible fashion and written in standard English?

Reviewer #1: Yes

Reviewer #2: Yes

6. Review Comments to the Author

Reviewer #1: To enhance the clarity and consistency of the manuscript, I recommend a few minor revisions. Firstly, consolidate the information regarding the company names of chemicals and kits used throughout the manuscript into a single section within the Materials and Methods, specifically where the phycocyanin materials are described. This will streamline the text and improve readability. Secondly, for the histological images, remove the freehand circles and only arrows are sufficient. Ensure each arrow is clearly labeled to indicate its significance, thus providing precise and comprehensible visual data. Lastly, correct the spelling error in the graph of Figure 2, specifically the term "muscle tension," and ensure that all syntactical elements are accurate. These adjustments will significantly enhance the overall quality and professionalism of the manuscript. Thank you to the authors for revising the manuscript as per the reviewer’s suggestions.

Response: Thank you for your recommendation, we agree with the revision following your guidelines.

1. We consolidated the information on all chemicals and kits used in the Materials and Methods.

2. We removed freehand circles and stars in the histological images and showed only arrows. We have clarified the arrow in the figure legend. The bold arrow ( ) represents muscle fiber, the arrow (→) represents the nucleus, and the arrowhead (▲) represents the perimysium.

3. We corrected the spelling error in Figure 2 and checked for the typo throughout the main manuscripts.

Reviewer #2: (No Response)

7. PLOS authors have the option to publish the peer review history of their article (what does this mean?). If published, this will include your full peer review and any attached files.

Do you want your identity to be public for this peer review? For information about this choice, including consent withdrawal, please see our Privacy Policy.

Reviewer #1: No

Reviewer #2: No

---

## [Editor Report · Decision Letter 2]

26 Aug 2024

Phycocyanin attenuates skeletal muscle damage and fatigue via modulation of Nrf2 and IRS-1/AKT/mTOR pathway in exercise-induced oxidative stress in rats

PONE-D-24-00127R2

Dear Dr. Tunsophon,

We’re pleased to inform you that your manuscript has been judged scientifically suitable for publication and will be formally accepted for publication once it meets all outstanding technical requirements.

Kind regards,

Keisuke Hitachi

Academic Editor

PLOS ONE
---

## [Editor Report · Acceptance letter]

30 Aug 2024

PONE-D-24-00127R2 

PLOS ONE

Dear Dr. Tunsophon, 

I'm pleased to inform you that your manuscript has been deemed suitable for publication in PLOS ONE. Congratulations! Your manuscript is now being handed over to our production team.

Kind regards, 

on behalf of

Dr. Keisuke Hitachi 

Academic Editor

PLOS ONE